# Hybrid Digital-Droplet Microfluidic Chip for Applications in Droplet Digital Nucleic Acid Amplification: Design, Fabrication and Characterization

**DOI:** 10.3390/s23104927

**Published:** 2023-05-20

**Authors:** Beatriz J. Coelho, Joana P. Neto, Bárbara Sieira, André T. Moura, Elvira Fortunato, Rodrigo Martins, Pedro V. Baptista, Rui Igreja, Hugo Águas

**Affiliations:** 1CENIMAT|i3N, Department of Materials Science, NOVA School of Science and Technology, Campus de Caparica, NOVA University of Lisbon and CEMOP/UNINOVA, 2829-516 Caparica, Portugal; 2UCIBIO, I4HB, Department of Life Sciences, NOVA School of Science and Technology, Campus de Caparica, NOVA University of Lisbon, 2829-516 Caparica, Portugal

**Keywords:** digital microfluidics, droplet microfluidics, flow focusing, negative pressure

## Abstract

Microfluidic-based platforms have become a hallmark for chemical and biological assays, empowering micro- and nano-reaction vessels. The fusion of microfluidic technologies (digital microfluidics, continuous-flow microfluidics, and droplet microfluidics, just to name a few) presents great potential for overcoming the inherent limitations of each approach, while also elevating their respective strengths. This work exploits the combination of digital microfluidics (DMF) and droplet microfluidics (DrMF) on a single substrate, where DMF enables droplet mixing and further acts as a controlled liquid supplier for a high-throughput nano-liter droplet generator. Droplet generation is performed at a flow-focusing region, operating on dual pressure: negative pressure applied to the aqueous phase and positive pressure applied to the oil phase. We evaluate the droplets produced with our hybrid DMF–DrMF devices in terms of droplet volume, speed, and production frequency and further compare them with standalone DrMF devices. Both types of devices enable customizable droplet production (various volumes and circulation speeds), yet hybrid DMF–DrMF devices yield more controlled droplet production while achieving throughputs that are similar to standalone DrMF devices. These hybrid devices enable the production of up to four droplets per second, which reach a maximum circulation speed close to 1540 µm/s and volumes as low as 0.5 nL.

## 1. Introduction

Microfluidic technologies used for handling low volumes of liquid have been an object of high interest over the past decades, and are particularly relevant for developing portable platforms with biotechnological applications [1,2]. Such interest is driven by the numerous advantages of microfluidic systems, which include low-volume reactions, portability, rapid prototyping, or high reaction throughput [3,4]. Several methodologies have been developed, namely paper microfluidics (PMF), digital microfluidics (DMF), continuous-flow microfluidics (CMF), and droplet microfluidics (DrMF) [5,6,7]. Natural evolution has triggered the fusion of different types of microfluidics to create synergies between the combined technologies [7]. Examples of hybrid approaches include paper and digital microfluidics [8], paper and continuous-flow microfluidics [9], and digital microfluidics and continuous-flow [10,11,12,13,14], or droplet microfluidics [15,16,17]. Specifically regarding the fusion between DMF and CMF, the CMF microfluidic channels (coupled with syringe pumps) are commonly used for reagent loading or withdrawal from DMF devices [10,11,12] or chemical separations [13,14]. Joining DMF and DrMF, on the other hand, has been applied to a variety of cell sorting and analysis protocols [15,16,17].

Herein, we explore single-substrate DMF–DrMF hybrid devices, where the DMF portion successfully supplies samples to a nano-droplet generator, which in turn produces droplets employing a pulling force, or negative fluidic pressure, applied by a syringe pump. Briefly, DMF involves individual droplet control by voltage actuation over a dielectric layer (thus effectively controlling the liquid input to DrMF), whereas DrMF takes advantage of the immiscibility between water-based liquid and oil to create nano-droplets at a flow-focusing region. The major objective of joining DMF and DrMF in the present work is to overcome some disadvantages of the separate technologies, particularly the difficulty of mixing different reagents due to the predominance of laminar flow in microfluidic channels (associated with a low Reynolds number), which is frequently highlighted as one of the main disadvantages of channel-based microfluidics [18,19]. Double-plate DMF technologies (where droplets flow sandwiched between two plates) are capable of solving the mixing issue since droplets may be voltage-actuated to perform mixing protocols ranging from a back-and-forth motion [20], circular motion [21], frequency-assisted mixing [22], or combinations of the latter [23]. Nevertheless, DMF technologies may be slightly more expensive than DrMF technologies in terms of the materials comprising the devices since DrMF devices may rely only on a silicone-based polymer and window glass, whereas DMF devices include several layers, which may contribute to the increase in the final price (e.g., hydrophobic layer or gold electrodes). Moreover, DrMF typically allows for lower-volume droplets than DMF, with a higher and faster droplet throughput. Thus, by uniting both technologies, we take advantage of the higher mixing efficiency of DMF and the higher throughput and lower volume droplets of DrMF to create a hybrid system, envisioning future applications to nucleic acid amplification and quantitative detection.

In previous works, we have produced DMF-only devices for nucleic acid amplification monitoring [23] and DrMF-only devices for endpoint nucleic acid detection [24]. Our DMF devices proved to be successful in reagent mixing, but presented low-throughput, whereas our DrMF devices did not enable reagent mixing, but presented very high-throughput. To enable mixing while improving throughput, we have combined both microfluidic technologies on a single substrate. We focus on DMF–DrMF hybrid devices operating on negative fluidic pressure and further evaluate the flow conditions required for producing on-demand nano-liter droplets of a common buffer in nucleic acid amplification. We assess the droplet volume, circulation speed, and production frequency of our hybrid devices, and compare these performance indicators with standalone DrMF devices.

A nucleic acid amplification buffer was chosen as a test sample considering the inarguable importance of such assays in diagnostics, from bacterial infections to cancer or even viruses, such as SARS-CoV-2. A polymerase chain reaction (PCR) is considered the gold standard for nucleic acid amplification [25,26], but recently, increasing attention has been directed toward isothermal amplification systems, such as loop-mediated isothermal amplification (LAMP), with comparable robustness and sensitivity [27,28,29]. Moreover, LAMP is compatible with DMF [23,30,31,32,33] and DrMF [24,34,35]. 

Using the developed DMF–DrMF platform, we demonstrate the generation of nano-liter droplets from a commercial buffer used in LAMP reactions, fed by DMF electrodes, and controlled by a combination of positive pressure (pushing flow) applied to the oil phase and negative pressure (pulling flow) applied to the aqueous phase (LAMP buffer). Negative pressure has been previously studied in droplet generation systems [36,37], and has also been used as a driving force for DMF–CMF hybrids [14]. However, little attention has been paid to how fluidic pump volumetric flow rates (referred to as flows for simplification), for both the oil and the water-based liquid phases, influence the droplets produced. Our hybrid devices allow for the continuous production of droplets at a rate of two to four droplets per second, which travel at high speeds (from 877 µm/s to 1540 µm/s) and present volumes below 2 nL. Other nano-droplet generation systems using negative pressure produce droplets with higher volumes [38], and do not enable on-chip reagent mixing prior to droplet formation [36,37,38,39,40], which would decrease the probability of human error and contamination issues. With the described platform, we envision mixing all the reagents required for nucleic acid amplification with the DMF section, performing the reaction by heating the platform and further increasing the output of the reaction to thousands of droplets through nano-liter droplet generation with the DrMF section. Temperature control may be performed through a previously disclosed system, and amplification detection may rely on a fluorescence detection module adapted to the fluorescent reporter chosen for the reaction. This setup will thus enable the quantification of the initial nucleic acid target by droplet digital LAMP or PCR, which is crucial in applications such as gene expression analysis, mutation detection, or the determination of viral loads [41,42].

## 2. Materials and Methods

The DMF–DrMF hybrid device follows an in-line approach, where both moieties of the device are built on the same standard glass substrate. Thus, sample fluid flows directly from the DMF section to its DrMF complement—Figure 1.

The design includes a DMF section with a bottom plate comprising three working electrodes and a larger reservoir (Figure 1b). An access port was drilled into the top plate, directly above this larger reservoir, thus enabling sample insertion. The final electrode was optimized to facilitate the transference of the aqueous phase from the DMF section to the DrMF section. This electrode features a triangular termination that directs the sample toward the DrMF section, and further includes a rectangular protuberance that enters the DrMF section, further assisting liquid transference (Figure 1b,c). The DrMF section consists of a standard flow-focusing device, where phase and flow rate differences allow for the generation of nano-liter droplets at the junction area. For details regarding DrMF and the specific dimensions of the droplet generator, please refer to Appendix A. The DMF–DrMF hybrid is mounted onto a custom 3D-printed casing, which enables communication with the voltage control system, as previously described [23]. Moreover, standalone DrMF devices were fabricated to optimize and evaluate droplet generators operating with a pulling force, and to assess the influence of passive liquid supply (i.e., not driven into the device by the assistance of a syringe pump). The architecture of the standalone DrMF devices is similar to that of hybrid devices; however, an additional squared reservoir (with approximately the same dimensions as the DMF electrodes) is used to store the liquid phase, which in turn is dispensed from a previously filled pipette tip—Figure 2.

Before performing any fabrication step, the glass substrates were subjected to a thorough cleaning process. Firstly, a common dishwashing detergent (an outstanding surfactant) was used to gently rub the substrates, which were then rinsed in ultrapure water and blow-dried with a nitrogen jet. Substrates were then cleaned with acetone followed by isopropanol in an ultrasound bath for 15 min each. Subsequently, substrates were rinsed with ultrapure water, blow-dried with a nitrogen jet, and further dried on a hot plate at 180 °C for 10 min to remove any remaining water molecules on the surface. After cleaning, the DMF section is first fabricated, followed by the DrMF section, as described below. Please note that some of the fabrication steps, such as the deposition of the dielectric layer (Parylene C), are common to both sections.

### 2.1. DrMF Device Fabrication

The DrMF section consists of a set of microchannels embossed on a PDMS (polydimethylsiloxane) substrate, which is then sealed to the glass substrate containing the DMF electrodes. Such PDMS-based microchannels were produced from primary SU-8 molds. Briefly, SU-8 2150 (Kayaku Advanced Materials, Westborough, MA, USA) was spin-coated at 500 rpm for 7 s and 3260 rpm for 30 s (considering a room temperature of 23 °C), after which the resist was allowed to rest for 10 min. Soft-baking followed, with heating at 65 °C for 5 min and 95 °C for 20 min. After a brief cooldown, the SU-8 resin was exposed on a mask aligner (Suss MA6 UV from Suss MicroTec SE, Garching, Germany) with a constant dose of 240 mJ·cm^−2^. After exposure, two additional baking steps were performed, with heating at 65 °C for 5 min and 95 °C for 10 min. Finally, the molds were developed in propylene glycol monomethyl ether acetate (Sigma-Aldrich, St. Louis, MO, USA) for 10 min on a rotation table rotating at 50 rpm. Such molds were then used to fabricate the PDMS microchannels. Firstly, base and curing agents from the SYLGARD^®^ 184 silicone elastomer kit (Dow, Midland, MI, USA) were mixed on a 10:1 proportion and deposited over the SU-8 molds, which were previously placed in a plastic Petri dish. The Petri dish was then placed in vacuum until all air bubbles were eliminated. Following this step, mold and elastomer were placed in a drying oven at 70 °C for approximately 1 h, after which the PDMS elastomer could be successfully peeled from the SU-8 mold. The photolithographic mask includes a guiding line at the interfacial region with DMF, which is transferred onto the PDMS and creates a groove. This groove allows for accurate cutting of the polymer without damaging the flow-focusing region at this point in fabrication. Access ports to the DrMF were then opened on the PDMS with an appropriate puncher (Darwin Microfluidics, Paris, France). Sealing of the PDMS microchannels onto the glass substrate followed by plasma exposure with 30 SCCM of O_2_ for 70 s at 1000 mTorr and 25 W (Phantom III RIE from Trion Technology, Tempe, AZ, USA). A total of 7 μL of isopropyl alcohol (IPA) was placed between the DrMF section and the glass substrate for alignment of the flow-focusing region with the interfacing electrode (see Figure 1c) of the DMF section under a microscope (the IPA creates a sacrificial layer that allows the user to adjust the DrMF section on the glass substrate if necessary). Another heating step followed, for approximately 2 h at 115 °C in a hot plate with added weight on top. The DrMF section is thus fabricated, and the remaining DMF fabrication steps are followed (see Section 2.2). Please note that this protocol was also used to fabricate standalone DrMF devices.

### 2.2. DMF–DrMF Device Fabrication

The fabrication of the DMF section begins with the patterning of the DMF electrodes onto the glass substrates. Positive photoresist (AZ ECI 3012 1.2 µm grade, from MicroChemicals, Ulm, Germany) was spin-coated on the substrates at 2000 rpm for 10 s, and 4000 rpm for 20 s for thickness definition. Following spin-coating, substrates were pre-baked on a hot plate at 115 °C for 75 s and exposed for 6.5 s on a mask aligner (Suss MA6 UV from Suss MicroTec SE, Garching, Germany). Post-baking followed at 115 °C for 30 s, after which samples were developed in AZ726 MIF developer (Micro-Chemicals, Ulm, Germany). A 200 nm layer of chromium was then deposited using a homemade electron beam deposition system, with substrate heating at 100 °C to improve adhesion, and the DMF electrode pattern was finally defined after lift-off with acetone. Following electrode patterning, the DrMF section was aligned with the connection electrode and sealed onto the glass substrate, and prior to the dielectric deposition step, the top of the DrMF section was covered with polyimide tape (Pro-Power Farnell, Leeds, UK). Covering the top of the DrMF section prevents the deposition of Parylene C on top of the microfluidic channel where droplets circulate, which creates a blurring effect that subsequently hinders video capture of the droplet shapes. For the dielectric layer, a 2 µm layer of Parylene C was deposited using a Labcoater^®^ PDS 2010 system (Specialty Coating Systems, Indianapolis, IN, USA) from Parylene C dimer provided by the same manufacturer. This deposition was performed in a double-layered format, with two 1 µm layers deposited sequentially for pinhole reduction. For each deposition, 3 drops (dispensed with a common dropper) of adhesion promoter (Silane A 174 from Merck KGaA, Darmstadt, Germany) were brushed onto the deposition chamber with a standard acrylic paint brush. Finally, a hydrophobic layer comprised of a 0.6% *wt*/*wt* solution of PTFE (polytetrafluoroethylene) AF 1600 in Fluorinert FC-40 (DuPont, Wilmington, DE, USA) was deposited by spin-coating at 1000 rpm for 30 s for a thickness of approximately 50 nm, and substrates were further baked at 160 °C for 10 min. Regarding the top plate, where the ground connection is established, commercially available ITO-coated glass was used, with holes drilled by a diamond drill tip for sample insertion. Prior to use, a PTFE-based hydrophobic layer was deposited over the top plates, as described above. Three strips of polyimide tape were used to create a 180 µm gap between the top and bottom plate, and finally, nail polish was placed around the electrode area for sealing and isolation (hot plate cure at 55 °C for 10 min). Before use, the DMF section was filled with silicone oil (5 cSt-Sigma-Aldrich, St. Louis, MO, USA), previously degassed in vacuum for a minimum of 3 h.

### 2.3. DMF–DrMF Experimental Setup

Two separate control systems were used to operate the hybrid DMF–DrMF devices, with one applied to each section. For the DMF section, a custom control system similar to the one previously disclosed by our group [23] was used. Briefly, electrode control is mediated by an Arduino Mega board, which is acted upon via an in-house built Python-based software (version 3.7.6). The Arduino board is in turn connected to a custom PCB (printed circuit board), which controls the voltage applied to the DMF electrodes meant to be actuated or non-actuated. The second control system was applied to the DrMF section, and was comprised of two syringe pumps (Legato 210 P from KD Scientific, Holliston, MA, USA): one used to push oil into the DrMF section, and another one to apply negative pressure to the system, thus effectively pulling the aqueous phase. It is likely that the contrary pressures will slightly influence each other; however, such influence did not hinder device operation. For both oil and aqueous phases, specific syringes were used (0.5 mL volume, model GASTIGHT^®^ 1750TLL, PTFE luer lock from Hamilton, Reno, NV, USA) coupled with microfluidic tubes (reference LVF-KTU-13 from Darwin Microfluidics, Paris, France) and blunt-end luer lock syringe needles (reference AE-23G-100x, also from Darwin Microfluidics). For standalone DrMF experiments, only the latter control system was used, with the aqueous phase being supplied by a pre-filled pipette tip instead of the DMF section, as previously stated. The oil phase circulating within the DrMF section consists of mineral oil (light oil for molecular biology, reference M5904 from Sigma-Aldrich, St. Louis, MO, USA) and Span80^®^ (Croda International PLC, Snaith, England) at 0.5% *wt*/*wt*, whereas the aqueous phase consists of LAMP buffer diluted to 1× (ThermoPol^®^ reaction buffer from New England Biolabs, Ipswich, MA, USA). As mentioned in Section 2.2, the oil circulating on the DMF section consists of 5 cSt silicone oil (Sigma-Aldrich, St. Louis, MO, USA). The DMF–DrMF devices and physical support were placed on a clear acrylic base, which was mounted on an optical breadboard (model MBH3060/M from ThorLabs, Newton, NJ, USA). Below the acrylic base, a custom-built lighting base with multiple LED lights allowed for optimal illumination and contrast control of the DrMF droplets for further image-based droplet analysis. Finally, a 25 mm optical construction rail (ThorLabs) was mounted onto the optical table to hold a Chameleon3 USB3 monochrome camera (model CM3-U3-13Y3M-CS, Python 1300 from Teledyne FLIR, Wilsonville, OR, USA), coupled with an MLM3X-MP magnification lens from Computar (Cary, NC, USA) to acquire videos.

### 2.4. Experimentation on DMF–DrMF Hybrid Devices

After mounting the experimental setup described in Section 2.3, silicone oil should be slowly introduced into the DMF section, thus filling the DMF cavity without producing air gaps. An additional hole may be drilled on the top plate to facilitate the exit of air between the DMF top and bottom plates as silicone oil is introduced. Mineral oil should be inserted into the DrMF section at a relatively low rate (500 nL/min), thus slowly expelling air from the device without the formation of air bubbles. Devices are only used after ensuring that no air bubbles are observed on either section. Despite some contact between the oils used in both moieties at the interface, they are immiscible, and this contact does not significantly affect the operation of the device. According to the principles of DMF, the LAMP buffer should then be inserted into the DMF section of hybrid devices by activating the larger reservoir and then inserting the buffer through the hole drilled on the top plate, by means of a micropipette. By sequentially activating and deactivating neighboring electrodes, the LAMP buffer should be moved toward the interfacing electrode (see Figure 1b,c), and only then should the respective syringe pump be activated (negative pressure, pulling force). For droplet motion in DMF, an AC signal of 40 V_RMS_ was used at a frequency of 5 kHz. This signal was kept constant at the interfacial electrode during droplet generation to ensure a constant supply of LAMP buffer from the DMF section to the DrMF section. Appendix A demonstrates a working DMF–DrMF hybrid device from sample insertion on the DMF section until droplet production at the flow-focusing region. Moreover, regarding the testing of different fluidic flows on the DrMF section, tests should always be performed from the smaller to the larger flows to avoid excessive pressure increases within the device. This increase in pressure could result in abnormal droplet formation, accompanied by larger volume and speed variability.

### 2.5. Data Acquisition and Analysis Workflow

Data acquisition and processing were fully centralized into an in-house-developed offline software. Firstly, a connection to the camera was established, and the acquired video (20 s to 25 s duration) was cropped to ensure that only the region where droplets flowed was captured. After cropping, the video frames were converted to grayscale to facilitate the definition of a threshold between the droplets and the surrounding medium. To achieve a fair droplet threshold, it is essential to adjust the LED lighting source to ensure that the edge of droplets is well defined, with a dark droplet contour and a bright droplet. Visual confirmation of the identified droplet contour may be performed by the user. After isolating the droplets from the background, the software determines the volume of droplets, the droplet’s speed, and the droplet’s production frequency. Finally, results obtained from the image processing may be exported to a .csv file for further analysis.

## 3. Results and Discussion

### 3.1. Standalone DrMF Devices

First, standalone DrMF devices were used to optimize the droplet generator design to determine the optimal conditions for producing high-throughput nano-droplets with negative pressure (pulling force). Regarding droplet production, two major parameters require optimization: the pulling flow for the LAMP buffer and the pushing flow for the oil phase. Several combinations of both were tested, and the droplet-forming conditions relied on pulling flows in the order of µL/min and pushing flows in the order of nL/min. Figure 3 illustrates the droplet volume, speed, and frequency obtained for several droplet-forming conditions (i.e., flow ratios), as well as a representative video frame of the droplets produced by the DrMF device. The droplet volume was determined from the diameter (d) measured by the image analysis software in two different conditions: if the droplet diameter is larger than the channel dimensions (height and width), the volume was determined as suggested in [43]; otherwise, the volume was approximated to the volume of a sphere with radius d/2. Please note that the frequency represents the number of droplets formed per second, denoted as dr/s for simplicity. The optimal flow for pulling the aqueous phase was determined to be 1000 nL/min. Below 1000 nL/min, droplets are not spherical as intended, but assume a rectangular shape with round corners. Above 1000 nL/min, droplet speed becomes too high and internal pressure increases enough to hinder the uniformity of the droplet production. Several flows for pushing the oil phase were further tested.

As illustrated by Figure 3, for each tested condition in both droplet volume and droplet speed, the data are narrowly distributed, suggesting that these standalone DrMF devices can generate stable droplets. Regarding the volume of the produced droplets, there is a decrease with increasing pushing flows. If we increase the oil flow at the flow-focusing region (Figure 1c) for a constant buffer flow, the buffer will be divided into smaller droplets (smaller volumes). Furthermore, this decrease in droplet volume is accompanied by an increase in droplet speed—Figure 3b. This may be explained by the increase in lateral pressure at the flow-focusing region as the pushing flow increases, which leads to faster droplet production and, consequently, faster droplet motion [45,46,47]. It is observable that the increase in droplet speed does not always increase linearly with the increase in oil flow. As a matter of fact, the increase in speed accompanying the flow change from 900 nL/min to 920 nL/min is almost as large as the one from 750 nL/min to 900 nL/min (the median (the median value was selected as a means of comparison since it is less affected by the outliers than the mean value) speed value increases approximately 204 µm/s and 173 µm/s, respectively). This tendency will be further explored in Section 3.3.

Droplet frequency (Figure 3c) is also a relevant parameter to measure, as it represents the number of droplets formed by the flow-focusing region per second. For DrMF standalone devices, an interesting behavior is observed: the frequency steadily increases as the oil flow increases until reaching a maximum for an oil flow of 920 nL/min. After this maximum, the droplet frequency decreases for higher oil flows. It was expected that higher oil flows would lead to a higher number of droplets produced per period, considering that the aqueous phase (LAMP buffer) will be disrupted at increasing rates. One possible explanation for the decrease in frequency after 920 nL/min of oil flow is the equilibrium of pulling and pushing forces acting upon the aqueous and oil phases, respectively. As the pushing oil flow approaches the pulling flow for the LAMP buffer (1000 nL/min), the opposite forces tend to balance with each other, and thus droplets are formed at an increasingly slower ratio. However, even though the frequency of droplet formation decreases, the droplet speed is kept stable. This suggests that it is possible to individually control the frequency of droplet formation, as well as the droplet speed.

### 3.2. DMF–DrMF Hybrid Devices

Knowing the flow combinations that lead to droplet formation studied with DrMF standalone devices, the same flow combinations were applied to DMF–DrMF hybrid devices. As mentioned in Section 2.4, in hybrid devices, the LAMP buffer must first be moved toward the interfacing electrode of the DMF section, and only then can the DrMF syringe pumps be activated for droplet formation. For all the oil flows tested on standalone DrMF devices, a LAMP buffer was successfully supplied by the DMF section to the DrMF section, where nano-liter droplets were further produced. This is a reasonable indicator that the union of both microfluidic moieties is at the very least able to match their standalone capabilities. Figure 4 illustrates the droplet volume, speed, and frequency obtained for the mentioned hybrid devices, as well as an example of the droplet-forming interface.

DMF–DrMF hybrid devices also produce stable droplets, as evidenced by the low dispersion of data in both droplet volume (Figure 4a) and speed (Figure 4b) for all the tested conditions. This attests to the quality and robustness of the developed devices. A deeper analysis of the volume shows that there is a clear tendency for a decrease as the oil flow is increased (see Section 3.3). Figure 4a reveals that the decrease in the average droplet volume from the first to the last tested oil flow condition (500 nL/min and 980 nL/min, respectively) is not very expressive. As a matter of fact, there is very little variation in the populations obtained for 500 nL/min and 750 nL/min (median droplet volumes of 1.5 nL and 1.4 nL, respectively), despite the large increase in oil flow. This might be due to the reduction in the microfluidic channel dimension at the interface between DMF and DrMF. Since both Parylene C and PTFE are deposited on the hybrid devices after bonding the DrMF section to the glass substrate (see Figure 1a), two supplementary layers of thickness are added to the walls of the PDMS. Appendix A includes a study on the variation of the microfluidic channel width as a result of Parylene C and PTFE depositions. This decrease in the diameter of the DrMF microfluidic channel at the flow-focusing region leads to a decrease in the maximum droplet volume when compared with standalone DrMF devices. Moreover, PTFE is even more hydrophobic than PDMS, reducing the interaction of the aqueous phase with DrMF channel walls, which could also contribute to the non-decrease in the droplet volume with the increase in oil flow. Appendix A includes information regarding the contact angle of the discussed materials as a measure of their respective hydrophobicity. The droplet volume only decreases for oil flows from 900 nL/min onward, as illustrated in Figure 4a. Regarding the droplet speed (Figure 4b), as discussed in Section 3.1, speed increases with increments in the oil flow. However, for DMF–DrMF hybrid devices, speed presents little variation between 500 nL/min and 750 nL/min of oil flow (1274.1 µm/s and 1212.6 µm/s median speed, respectively), and even between 900 nL/min and 920 nL/min (1258.3 µm/s and 1378.8 µm/s median speed, respectively). This could be explained by the activation of the interfacing electrode during the droplet generation process, which could exert an additional force on the droplets, thus attracting them to the activated DMF electrode. Such additional force could act as a barrier to the increase in droplet speed for sequentially larger oil flows without influencing the droplet volume. Regarding droplet frequency (Figure 4c), the number of droplets produced per second increases as the oil flow is increased until 920 nL/min. After such oil flow, the droplet frequency decreases, which could also be explained by the equilibrium between pushing and pulling forces, as the pushing oil flow becomes closer to the pulling flow of the LAMP buffer (1000 nL/min). However, this decrease in droplet production does not affect droplet speed, which continues to increase after 920 nL/min of oil flow. The data thus suggests that the hybrid DMF–DrMF devices enable on-demand nano-droplet formation with tunable droplet production rate and speed. However, it is observable that there is a higher dispersion of values than for standalone devices, which will be further analyzed in the following section.

### 3.3. Comparison between Standalone DrMF and Hybrid DMF–DrMF Devices

After individually studying the droplet parameters of standalone DrMF and hybrid DMF–DrMF devices, a comparison between both was performed. Figure 5 contains graphical representations of the average droplet volume, speed, and droplet production frequency for each tested oil flow.

As depicted in Figure 5a, there is a significant difference in the droplet volume between standalone and hybrid devices for 500 nL/min of oil flow. This difference decreases with the increase in oil flow; however, it remains statistically significant until the final tested oil flow, 980 nL/min. This difference in droplet volume between both types of devices can be explained by the thinning of the DrMF microfluidic channel in DMF–DrMF hybrid devices, due to the deposition of Parylene C and PTFE (see Appendix A). Standalone DrMF devices present a clear decrease in droplet volume (almost linear) as the oil flow increases until 900 nL/min. From this point forward, the linear tendency appears to be lost; however, lower oil flows continue to produce smaller droplets. As for hybrid devices, there is no linear tendency for droplet volumes, which again, could be related to the decrease in the microfluidic channel diameter. From 900 nL/min to 980 nL/min, the difference in the average droplet volume between standalone and hybrid devices is less pronounced; however, there is still a clear distinction in the behavior of the two types of devices. In this oil flow regime, the decrease in volume is smoother in hybrid than in standalone devices, which is probably due to the difference in the initial volume. As for Figure 5b, in standalone DrMF devices, there is a linear tendency for the droplet speed to increase as the oil flow increases, until 900 nL/min. After 900 nL/min flows, the linear tendency appears to be lost and is replaced by what seems to be a saturation plateau tending toward approximately 1900 µm/min. For hybrid DMF–DrMF devices, the average droplet speed remains relatively stable for 500 nL/min and 700 nL/min, increasing only after the latter oil flow. This could be due to the force exerted by the activated interfacial electrode on the LAMP buffer, as previously stated (Section 3.2). Moreover, the droplet speed in hybrid devices is consistently lower than in standalone DrMF devices. This suggests that despite the presence of PTFE at the beginning of the DrMF microfluidic channel (which, in the form of the solution prepared for this work, presents a higher degree of hydrophobicity than PDMS alone—see Appendix A, the force exerted by the activated interfacial electrode is dominant, leading to the overall lower droplet speeds. Additionally, regarding hybrid devices, the speed does not seem to reach any saturation plateau; however, the maximum average speed achieved in such devices is still far from the plateau where standalone devices appear to saturate. As illustrated in Figure 5c, the frequency in standalone DrMF devices presents a closely linear increase with the increase in oil flow, from 500 nL/min to 900 nL/min. For higher oil flows, consistently with droplet volume and speed, the linear tendency is lost, and the maximum droplet production frequency is achieved for 920 nL/min oil flow. After reaching this maximum, the droplet frequency decreases, which, as previously stated (Section 3.1), is likely related to the equilibrium of the positive and negative pressures applied to the oil and LAMP buffer, respectively. Moreover, it is interesting to note that, contrary to droplet volume and speed, droplet production frequency is similar in both standalone and hybrid devices. DMF–DrMF devices thus match the high-throughput production of droplets achieved by standalone DrMF devices, yet with a slower progression of speed as the oil flow increases. This implies that hybrid devices provide the same droplet throughput with a more controlled progression of droplet volume and speed since such characteristics do not fluctuate as much as for standalone DrMF devices. In both cases, the maximum droplet production rate is achieved for 920 nL/min of oil flow. Furthermore, the frequency in DMF–DrMF devices presents a higher variability than of DrMF devices, as suggested by the larger standard deviation. One possible explanation is the activation of the interface electrode. The activation signal could be sufficient to introduce variations in the number of droplets produced per time unit. Nevertheless, regardless of the type of device tested, droplet volume, droplet speed, and droplet frequency can be successfully controlled by simply adjusting the oil flow.

## 4. Conclusions

Combined microfluidic technologies have the potential to overcome the limitations of each technology while elevating their respective advantages. In this work, we present a hybrid DMF–DrMF device where the DMF section is capable of mixing and further supplies the DrMF section with an on-demand LAMP buffer, and the DrMF section successfully generates nano-droplets through flow focusing. Particularly, the flow-focusing region operates with negative pressure applied to the outlet and positive pressure applied to the oil phase. We compared the droplets produced by standalone DrMF devices with hybrid DMF–DrMF devices, and after our analysis, we concluded that both methodologies provide controllable droplet production. The standalone DrMF devices provide a wider range of droplet sizes and circulation speeds, whereas DMF–DrMF devices provide a stabler and more controlled droplet production—both technologies provide similar droplet throughputs. This work presents a proof-of-concept for the working principles of hybrid DMF–DrMF devices; however, it is important to consider that DMF enables a wide range of electrode pathways and, consequently, a wide range of protocols, and DrMF further enables high-throughput nano-droplet production, which may flow in any desired microfluidic channel path. The fabrication of the hybrid DMF–DrMF devices is fully flexible, and numerous designs are possible for both sections. The work presented herein thus opens the path to a new range of applications featuring nucleic acid amplification, where the DMF section may be used to mix the required reagents (see Appendix A) and further supply the mix to the DrMF section, which is then able to produce nano-liter droplets, which are required for digital droplet reactions such as droplet digital LAMP [24] or droplet digital PCR [48]. By adding a heating element, a fluorescence detection system, and software for droplet analysis, we envision DNA or RNA quantification by in-line droplet-digital LAMP. This quantification will in turn empower applications such as viral load determination, analysis of gene expression, or detection of mutations. Beyond the field of nucleic acid amplification, our devices may be used for automatization of other applications, namely chemical synthesis (e.g., nano-particle synthesis), cell culture, cell sorting, or drug screening.

## Figures and Tables

**Figure 1 sensors-23-04927-f001:**
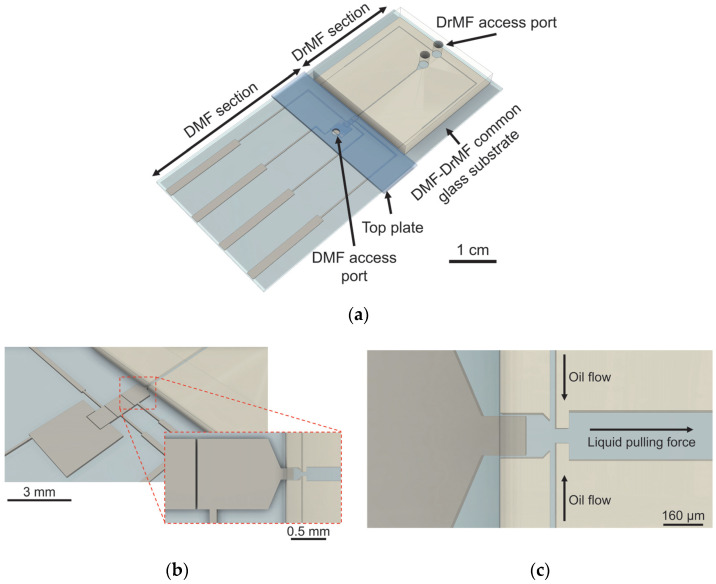
Three-dimensional illustration of a DMF–DrMF hybrid device. (**a**) General view of the hybrid device, evidencing all constituting parts. (**b**) Close-up view of the DMF electrodes without the top plate. The inset image represents the electrode specifically designed for the DMF–DrMF interface, which includes a triangular end shape that directs droplets toward the droplet generation site within the DrMF section. An additional protuberance enters the DrMF cavity and further assists the liquid transference from the DMF to DrMF. (**c**) Interface between DMF and DrMF, evidencing the flow-focusing area (droplet generator structure), as well as the direction of the oil phase flow and the flow direction of the droplets. Briefly, the oil phase is inserted through the access port closest to the edge of the DrMF section and flows bilaterally to the flow-focusing region, whereas the aqueous phase (LAMP buffer) is pulled from the DMF section through the second (inward) access port. Both pulling of the aqueous phase and insertion of the oil phase are mediated by syringe pumps.

**Figure 2 sensors-23-04927-f002:**
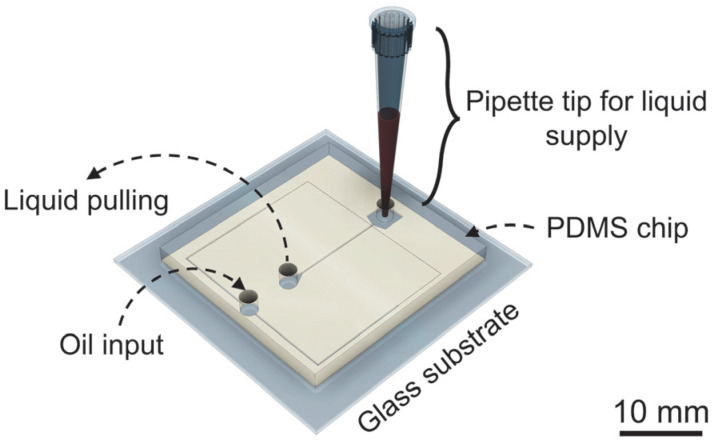
Three-dimensional illustration of a standalone DrMF device. On this standalone device, the oil phase is inserted through the oil input through a syringe pump (pushing force), the liquid is supplied by a pre-filled pipette tip connected to an additional squared reservoir, and a pulling force is applied through the outlet.

**Figure 3 sensors-23-04927-f003:**
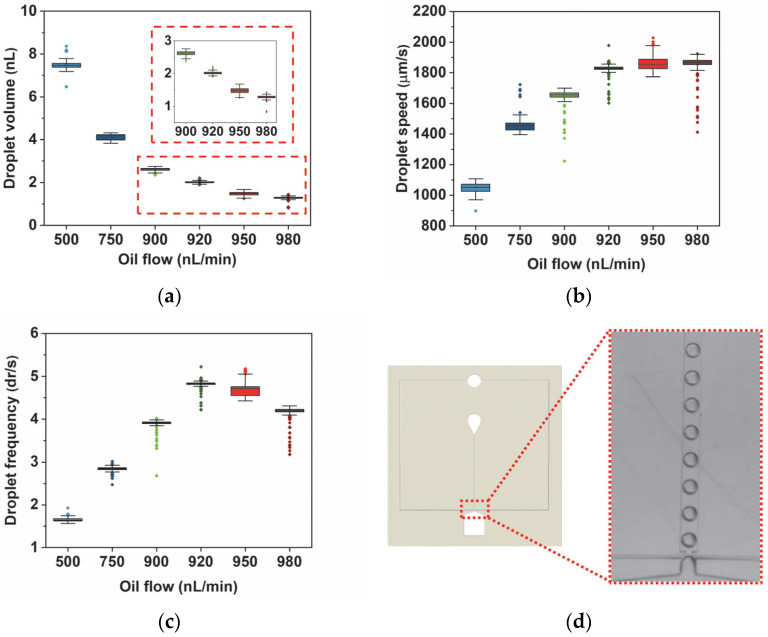
Box plots for droplet volume (**a**), speed (**b**), and frequency (**c**), as determined by the software developed for this work. The pulling flow for the LAMP buffer was kept constant at 1000 nL/min, whereas the pushing flow for oil varied from a minimum of 500 nL/min to a maximum of 980 nL/min. Below and above this range of pushing flows, droplet formation was unstable. Outliers were determined by the interquartile range (IQR) method [44] and kept in the data presentation, represented by colored points in line with the respective data set. For each experiment, a minimum of 1000 droplets were evaluated. (**d**) represents a standalone DrMF device, where the inset shows a video frame evidencing droplets produced with standalone DrMF devices, for a randomly selected oil flow of 920 nL/min.

**Figure 4 sensors-23-04927-f004:**
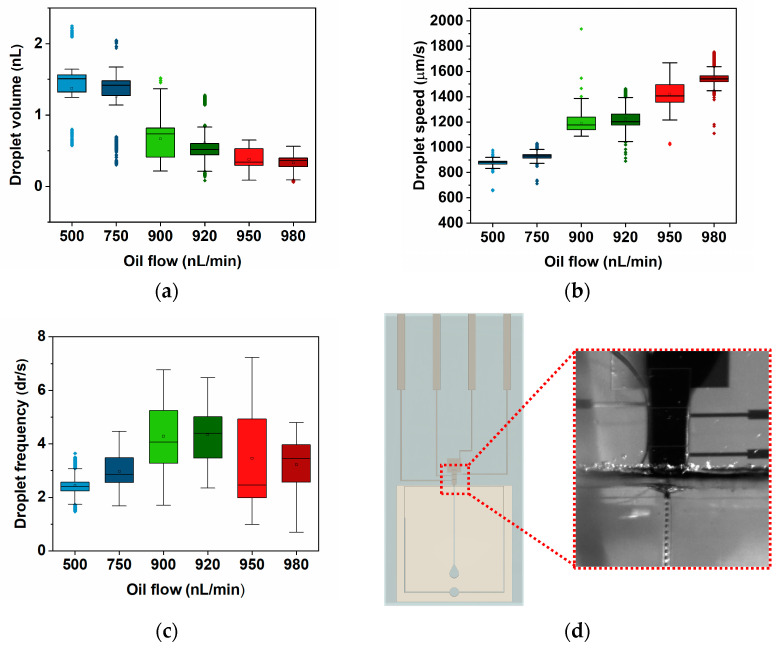
Box plots for droplet volume (**a**), speed (**b**), and frequency (**c**) in DMF–DrMF hybrid devices, as determined by the software developed for this work. As experimented for DrMF standalone devices, the pulling flow for the LAMP buffer was kept constant at 1000 nL/min whereas the pushing flow for oil varied from a minimum of 500 nL/min to a maximum of 980 nL/min. Once more, outliers were determined by the interquartile range (IQR) method [44] and kept in the data presentation, and represented by colored points in line with the respective data set. For each experiment, a minimum of 1000 droplets were evaluated. (**d**) represents a video frame evidencing droplets produced with hybrid DMF–DrMF devices for a randomly selected oil flow of 920 nL/min. LAMP buffer is supplied by the DMF section on top, droplets are formed on the near-interfacial flow-focusing region (not visible), and further flow through the DrMF microfluidic channel on the bottom.

**Figure 5 sensors-23-04927-f005:**
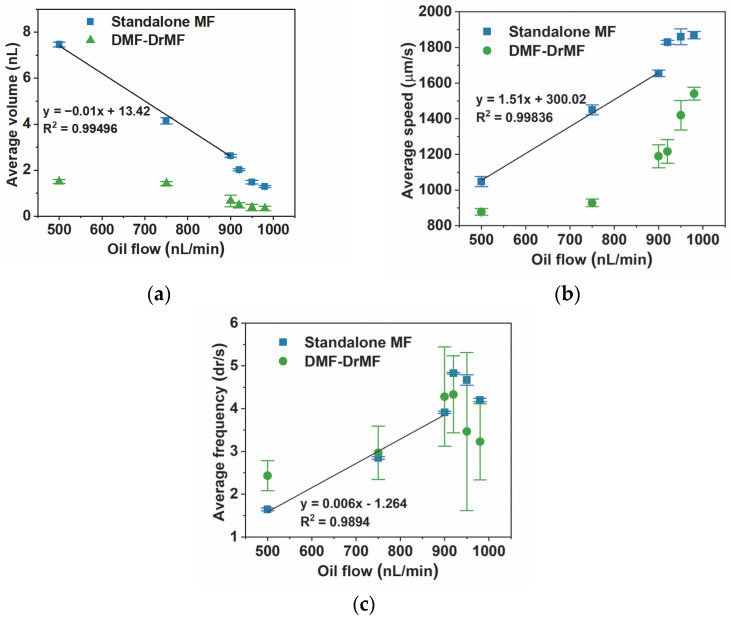
Average droplet volume (**a**), speed (**b**), and frequency (**c**) for variable positive pressure oil flows and constant negative pressure for LAMP buffer (1000 nL/min) in standalone DrMF devices and hybrid DMF–DrMF devices. Outliers were removed from all data sets, considering their large influence on the averaged results. The inset on (**a**) represents an enlarged plot of the area highlighted in red (between 880 nL/min and 1000 nL/min oil flows). In (**a**–**c**), the equations correspond to the linear fit obtained for the first three oil flow conditions (500 nL/min, 750 nL/min, and 900 nL/min) for standalone DrMF devices, also represented by the black lines. Error bars correspond to the standard deviation of data.

## Data Availability

The data presented in this study are available on request from the corresponding author.

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
