# Peer review of "Hybrid Digital-Droplet Microfluidic Chip for Applications in Droplet Digital Nucleic Acid Amplification: Design, Fabrication and Characterization"

_sensors, 2023, doi:10.3390/s23104927_

Round 1

Reviewer 1 Report

While the separate microfluidic technologies present unique feature and also some inherent drawbacks, the combination of several microfluidic technologies seems to be a good choice. This work  exploits the combination of digital microfluidics (DMF) and droplet microfluidics (DrMF) on a single substrate, where the authors take advantage of the higher mixing efficiency in DMF and the higher throughput and lower volume droplets of DrMF. Generally, this manuscript is well organized in structure and the academic expression is clear. However, a few questions need to be answered before the manuscript can be considered for further publication.

1. The title states that the hybrid microfluidic system is designed for qPCR tests. However, only digital LAMP (dLAMP) is emphasized in the contents. 

2. It is noticed that the authors have done a certain number of related studies before. Thus, it is necessary to state the novelty or difference of this manuscript when compared to previous publications.

3. It is better to present subfigures, like Figure 3d and Figure 4d, more clearly with more graphic illustrations.

4. In Figure 5c, the error bar seems to be too wider than a reasonable range.

Reviewer 2 Report

This manuscript reported Hybrid digital-droplet microfluidic chip for dPCR applications: design, fabrication and characterization. This work provides an idea about microfluidic chip for dPCR applications. However, this manuscript needs major revisions and the following comments will help to increase the quality of the manuscript before publication.

· Abstract should be more specific based on the findings

· In the "Introduction" section, general description on the importance of manuscript topic is poor. Therefore, the importance of this work cannot be well recognized from general readers. In order to fix this problem, addition of description on recent development in the field of research topic with citing recent comprehensive papers would be important. https://doi.org/10.3390/ijms23147957;

· How is the work different or better than those reported earlier? Author needs to highlight this in the introduction and discussion part.

· On what basis the oligonucleotides are selected and briefly explain in the text.

· Extensive editing of English language and style required

· Need to check consistency in reference the in text 

· Author should acknowledge the recommendation of present study in the conclusion section.

Extensive editing of English language and style required

Reviewer 3 Report

In this paper, digital microfluidics (DMF) and droplet microfluidics (DrMF) were combined on a single substrate.

It is an impressive work, but a revision is needed before accept:

1.        What is the effect of this system in the actual detection application? It seems like that this work only provide a technique for droplet generation. Temperature control and signal reading modules are required for complete detection system of PCR or LAMP. Or is it possible to combine this technique with existing detection system?

2.        Since DMF has higher mixing efficiency and DrMF has higher throughput and lower volume droplets, how is the mixing efficiency of the newly established method compared with DMF? How is the throughput and droplet volume of the newly established method compared to DrMF?

Reviewer 4 Report

Dear authors,

The publication is of course interesting, but it deals with a topic that has been covered many times:

1)10.3390/mi9050248

2) https://doi.org/10.1007/s41981-018-0025-2

The first paper seems to be similar to your paper. Both refer to pre-application microdroplets for specific purposes. publication (2) already shows a concrete application. That's why I think the introduction needs a bit of expansion to emphasize your work purpose.

further comments below:

1. line 113 - 116: This specific design was previously optimized by simulations performed with the COMSOL Multiphysics® software to produce droplets with volumes in the [1-10] nL range, necessary to perform digital PCR (dPCR) [38] or digital LAMP (dLAMP) [37] on-chip. - My question is where can we see these Comsol simulations? I thin this should be paced in the main manuscript.

2. Figure 2 is not needed. It can/should be linked to Figure 1. Furthermore, Figures 1(a) and 2 do not seem to match. As I understand it, Figure 1(a,b,c) are enlargements of Figure 2. If yes can you make it easier to follow.

3. at what temperature were the experiments carried out? The viscosity of the oil is strongly dependent on the temperature.

4. What is the viscosity of LAMP buffer?

The work is aesthetic and well written.

Round 2

Reviewer 2 Report

The author responded to all my comments and the editor can accept the revised version into appropriate journals. 

The author responded to all my comments and the editor can accept the revised version into appropriate journals. 

Reviewer 4 Report

Thanks for the clarifications and changes made. The work after the corrections is better. Therefore, I recommend it for publication.